# Is Unplanned Excision of Soft Tissue Sarcomas Associated with Worse Oncological Outcomes?—A Systematic Review and Meta-Analysis

**DOI:** 10.3390/cancers16020443

**Published:** 2024-01-19

**Authors:** Felipe Larios, Marcos R. Gonzalez, Kim Ruiz-Arellanos, George Aquilino E Silva, Juan Pretell-Mazzini

**Affiliations:** 1Facultad de Medicina, Universidad Peruana Cayetano Heredia, Lima 15102, Peru; felipe.larios.m@upch.pe (F.L.); kim.ruiz.a@upch.pe (K.R.-A.); 2Division of Orthopaedic Oncology, Department of Orthopaedic Surgery, Massachusetts General Hospital, Harvard Medical School, Boston, MA 02115, USA; mgonzalez52@mgh.harvard.edu; 3College of Arts and Sciences, Boston University, Boston, MA 02215, USA; georgeaquilinoesilva@gmail.com; 4Miami Cancer Institute, Division of Orthopedic Oncology, Baptist Health System South Florida, Plantation, FL 33324, USA

**Keywords:** unplanned excision, soft tissue sarcoma, local recurrence, residual disease

## Abstract

**Simple Summary:**

Unplanned excision (UE) of soft tissue sarcomas represents an important issue when treating this group of rare malignancies. It has been reported that UE may result in a poorer prognosis in terms of recurrence, survival, and other factors such as the need for amputation and plastic reconstruction procedures. Through a systematic review and meta-analysis of the available evidence, we aimed to answer some of the more relevant questions regarding oncological outcomes in these patients. Our study is the largest of its kind, and includes several important studies published over the last 5 years. We identified an association between unplanned excision and local recurrence, with special consideration to the impact of residual disease after an unplanned excision on local recurrence, and an association of local recurrence and worse overall survival in soft tissue sarcoma patients. Orthopaedic surgeons should consider re-excision as the standard approach when dealing with unplanned excision.

**Abstract:**

Background: Soft tissue sarcomas are a group of rare neoplasms which can be mistaken for benign masses and be excised in a non-oncologic fashion (unplanned excision). Whether unplanned excision (UE) is associated with worse outcomes is highly debated due to conflicting evidence. Methods: We performed a systematic review and meta-analysis following PRISMA guidelines. Main outcomes analyzed were five-year overall survival (OS), five-year local recurrence-free survival (LRFS), amputation rate and plastic reconstruction surgery rate. Risk ratios were used to compare outcomes between patients treated with planned and unplanned excision. Results: We included 16,946 patients with STS, 6017 (35.5%) with UE. UE was associated with worse five-year LRFS (RR 1.35, *p* = 0.019). Residual tumor on the tumor bed was associated with lower five-year LRFS (RR = 2.59, *p* < 0.001). Local recurrence was associated with worse five-year OS (RR = 1.82, *p* < 0.001). UE was not associated with a worse five-year OS (RR = 0.90, *p* = 0.16), higher amputation rate (RR = 0.77, *p* = 0.134), or a worse plastic reconstruction surgery rate (RR = 1.25, *p* = 0.244). Conclusions: Unplanned excision of Soft Tissue Sarcomas and the presence of disease in tumor bed after one were associated with worse five-year LRFS. Tumor bed excision should remain the standard approach, with special consideration to the presence of residual disease.

## 1. Introduction

Soft tissue sarcomas (STS) accounted for less than 1% of all malignant tumor diagnoses in the United States in 2023 [1]. Due to their asymptomatic nature and unspecific clinical presentation, these tumors are often mistaken for benign masses. [2,3,4]. Under the assumption of being benign, up to 53% of soft tissue sarcomas might be excised in a non-oncologic fashion without any regards for surgical margins or (neo)-adjuvant therapy [5,6].

Planned reoperation following an unplanned excision (UE) of a soft tissue sarcoma was first described by Giuliano et al. in 1985 [7]. They defined UE as the macroscopic removal of malignant lesions without consideration for preoperative imaging, biopsies, or adequate margins. In cases with UE, Giuliano et al. recommended definitive treatment with reoperation, either with or without radiotherapy. Recent studies have reported worse outcomes in patients with STS and a history of UE [8,9,10]. Consequently, the current standard of care for individuals who have undergone UE involves tumor bed excision with or without neo-adjuvant therapy. However, there is conflicting evidence on whether tumor bed excision is linked to higher local control rates, and whether this is associated with worse overall survival. Moreover, there is no consensus among surgeons on guidelines for tumor bed excision, especially with regard to timing and regimens for (neo)-adjuvant treatment.

In this setting, we aimed to answer the following questions: (1) Is unplanned excision associated with worse five-year overall survival and/or local recurrence-free survival? (2) Is residual disease on the tumor bed associated with worse five-year local recurrence-free survival? (3) Is local recurrence associated with worse five-year overall survival? (4) Is unplanned excision associated with a higher rate of amputation and/or plastic reconstruction surgery?

## 2. Materials and Methods

We performed a systematic review and meta-analysis following the Preferred Reporting Items for Systematic Reviews and Meta-Analyses (PRISMA) guidelines. The protocol was registered on PROSPERO (CRD42023437997).

### 2.1. Search Strategy

We performed a comprehensive search of the PubMed and Embase databases from inception to 30 June 2023. Our search strategy was conducted utilizing the following terms: (“sarcoma” [MeSH Terms] OR “Soft Tissue Neoplasms” [Mesh] OR “sarcoma” OR “soft tissue sarcom*” OR “soft-tissue sarcom*”) AND (“re-excision*” OR “reexcision*” OR “re-resection*” OR “reresection*” OR (“unplanned” AND “excision”)). Additionally, references of the included studies were manually screened to identify potential studies that had been missed in our search. 

### 2.2. Eligibility Criteria

Manuscripts that reviewed the following outcomes after re-excision of an unplanned excision of soft-tissue sarcomas of the extremities or trunk were included: five-year overall survival (OS), five-year local recurrence-free survival (LRFS), amputation rate (AMP), plastic reconstruction surgery rate (PRS) To analyze the association between unplanned excision and all outcomes mentioned previously, we included articles that used a comparison group (planned excision (PE) group).

To assess whether residual disease at re-excision after an unplanned excision was associated with local recurrence, we included articles that reported five-year LRFS based on the presence of tumor at definitive surgery in patients with a previous UE. Since no re-excision is performed for planned procedures, a comparison group was not required for these articles.

Finally, to assess whether local recurrence was associated with worse five-year OS, we also included articles that reported five-year OS based on the presence of local recurrence in patients with soft tissue sarcomas, whether they were subject to PE or UE. We did not require a comparison group for these studies.

The following exclusion criteria were applied: (1) case reports, case series and other nonpeer-reviewed publications such as conference proceedings and preprint servers, (2) data not displayed or not allowing for risk ratio and confidence interval calculations, (3) nonhuman studies and (4) articles that were not written in either English, Spanish, Portuguese or German. No additional filters on patient age, and/or year of publication were applied.

### 2.3. Selection, Data Collection and Extraction Process

A search query was separately conducted in each of the two databases employed and the resulting datasets exported to Covidence™ (Veritas Health Information, Deerfield, IL, USA). After removing duplicates, two reviewers (FL, KRA) independently screened studies for eligibility. In case of disagreement, the senior author (JPM) was consulted to reach a consensus.

Data was extracted in a standardized sheet. For qualitative data synthesis, the following variables were collected: institution(s) where the study was performed, year of publication, patient age, follow-up time, gender distribution, tumor depth, location, size, grade, most common histology, the use of adjuvant or neo-adjuvant radiotherapy and chemotherapy, definitive surgery margins, residual disease at re-excision and outcomes reported. For continuous variables such as age and follow-up time we collected mean or median values, according to how data was reported. For our quantitative analysis, we collected the number of events for a certain outcome in each group (patients with PE and UE) and the total amount of patients at risk in each group. Data was then displayed in contingency tables for statistical analysis.

### 2.4. Study Selection and Characteristics

Our search strategy resulted in 462 and 646 articles from PubMed and Embase, respectively (Figure 1). After removal of 384 duplicates, and with the addition of ten articles that were manually screened, titles and abstracts of 734 unique studies were screened and 664 of them were excluded. A total of 70 manuscripts were assessed for eligibility criteria and 43 studies met our eligibility criteria, and were finally included in our analysis [3,4,8,9,10,11,12,13,14,15,16,17,18,19,20,21,22,23,24,25,26,27,28,29,30,31,32,33,34,35,36,37,38,39,40,41,42,43,44,45,46,47,48].

We included 16,946 patients with soft tissue sarcomas, out of which 6017 (35.5%) had a previous UE. Presence of residual disease on the tumor bed after re-excision was reported in 1429 patients, and 850 (59%) had a positive result. The number of patients included in each study ranged from 14 [34] to 3913 [14] patients. All articles were retrospective cohort studies published in English between 1987 and 2023. The reported median or mean age in most articles was in the fifth and sixth decades of life (Table 1). Out of the 33 studies that had PE and UE groups, 26 reported information on tumor size. All 26 reported a higher mean or median in the PE group. Out of 24 studies that reported information on tumor depth, 22 reported a higher percentage of deep tumors in the PE group (Table 2). Adjuvant radiotherapy and chemotherapy were the most reported ways of oncologic treatment (Appendix A). The most common location was the lower extremity, followed by the upper extremity and trunk. The most common histologies were undifferentiated pleomorphic sarcoma, liposarcoma and synovial sarcoma (Appendix A).

Twenty-two studies including 6150 patients with PE and 3791 patients with UE were included to compare five-year OS outcomes [4,8,10,12,14,15,18,21,23,24,25,26,30,31,32,34,36,39,42,43,44,46]. Twenty-one studies, including 5182 patients with PE and 2204 patients with UE reported five-year LRFS [8,10,11,12,16,17,24,25,28,29,30,31,32,35,37,39,40,42,43,44,47]. Twelve studies, including 5393 patients with PE and 3290 patients with UE were included to compare AMP rates [4,14,16,17,21,24,25,35,39,40,42,44]. Information on PRS rate was available in twelve articles totaling 5488 patients with PE and 3263 with UE [4,9,12,14,20,21,24,30,36,39,42,44]. Twelve articles with 1429 UE patients reported residual disease status at re-excision, as well as the five-year LRFS [3,13,16,24,27,29,32,33,35,38,41,44]. Five articles with 1096 soft tissue sarcoma patients were included to compare five-year OS by local recurrence [8,19,22,45,48] (Table 3).

### 2.5. Assessment of Study Quality Assessment and Risk of Bias

Two reviewers (FL, GA) independently conducted quality assessments using the ROBINS-I tool. This tool is recommended by the Cochrane Collaboration and is used to assess risk of bias in non-randomized (observational) studies of interventions. All 33 studies that compared outcomes between planned and unplanned excision groups were subject to this assessment. The evaluation of each risk of bias domain was performed using the robvis online visualization tool. We analyzed thirty-three studies included and found all of them to fall into the category of ‘moderate’ risk of bias (Appendix A).

For the 10 articles that did not use a comparator group, we used the Strengthening the Reporting of Observational Studies in Epidemiology (STROBE) checklist field in accordance with previous orthopaedic literature [49,50,51]. For this checklist, 10 of the 22 items were used. Each item was graded on a scale of 0 to 2, with 2 points being used in well-described items, one point in partly described items, and 0 points in poorly described items. Studies with a cumulative score of ≥15 points were included. All 10 articles with no comparator group had scores above this threshold, and therefore were included (Appendix A).

### 2.6. Statistical Analysis

We used risk ratios to compare outcomes between the retrospective cohort studies found in each of the articles. Heterogeneity among studies was calculated using the I2-statistic. Values of I2 > 50% or *p* < 0.05 indicated significant heterogeneity between studies, and a random-effects model was chosen; otherwise, a fixed-effects (weighted with inverse variance) was used.

For meta-analysis results, we also calculated the fragility index and the ratio between the fragility index and the total number of participants for each significant outcome. The fragility index refers to the lowest amount of patients from any number of studies included in the meta-analysis for whom a change in their event-status (changing an event to nonevent or vice versa) would make the results no longer statistically significant (*p* > 0.05) [52]. All statistical analyses were performed with StataSE 14 (StataCorp, College Station, TX, USA).

### 2.7. Publication Bias

Publication bias was assessed using funnel plot analysis, in which the event rate is plotted against the inverse of the standard error (Appendix A). A Funnel plot was conducted for each analysis performed. Moderate asymmetry was found in all analysis, meaning some publication bias may be present.

## 3. Results


*(1)* 
*Is unplanned excision associated with worse five-year overall survival and/or local recurrence-free survival?*



Results showed that the presence of a UE was not associated with an increased risk of overall death at five years (RR 0.90 [95% CI 0.78 to 1.04], *p* = 0.16) (Figure 2). Having a UE was associated with a higher risk of local recurrence (RR 1.35 [95% CI 1.05 to 1.73], *p* = 0.019) (Figure 3A). A modification of event-status (i.e., changing an event to nonevent or a nonevent to event) in 3 patients (0.1% of the sample) would be necessary to make the effect of UE on five-year LRFS no longer statistically significant.


*(2)* 
*Is residual disease on the tumor bed associated with worse five-year local recurrence-free survival?*



A positive specimen at re-excision was associated with a higher rate of local recurrence at five years (RR 2.59 [95% CI 1.91 to 3.50], *p* < 0.001) (Figure 3B). A modification of event status in 14 patients (0.97% of the sample) would be necessary to make the effect of residual tumor positivity on five-year LRFS no longer statistically significant.


*(3)* 
*Is local recurrence associated with worse five-year overall survival?*



Local recurrence in STS patients was associated with a lower OS at five years (RR 1.82 [95% CI 1.35 to 2.46], *p* < 0.001) (Figure 4). The fragility index and ratio were 13 and 1.2% (13 of 1096), respectively.


*(4)* 
*Is unplanned excision associated with a higher rate of amputation and/or plastic reconstruction surgery?*



UE was not associated with a higher amputation rate in patients with soft tissue sarcomas (RR 0.77 [95% CI 0.54 to 1.08]) (Figure 5A). Similarly, UE was not associated with a higher rate of plastic reconstruction surgery procedures (RR 1.25 [95% CI 0.86 to 1.82]) (Figure 5B).

## 4. Discussion

Soft tissue sarcomas are often misdiagnosed as benign masses and are excised without proper oncological workup. The impact of these unplanned excisions on definitive treatment and patient outcomes is still debated and information is conflicting [6]. Our study demonstrated that UE and the presence of tumor at re-excision were associated with worse five-year LRFS. Although UE per se was not associated with worse survival, developing local recurrence was associated with worse five-year OS. We found no difference in amputation or plastic reconstruction surgery rates between patients with and without prior UE. While a previous systematic review [53] has been published on the topic, the authors did not perform any risk of bias assessment and missed several important studies published in the last 5 years [8,9,10,14,21,23,26,29,30,31,39,41,42,47]. With a total of 16,946 patients, 6017 of them with a previous unplanned excision, our study has a 62% larger sample and comprehensively assesses the biases in the included literature.

The results of this study are not without limitations. First, our analysis focused on the association between a previous UE and oncologic outcomes and did not control for potential confounders between groups, such as patient and tumor characteristics. In most articles, tumors in the PE group were larger and deeper in comparison with those in the UE group, which could potentially affect the results reported in each study. Second, given the observational nature of all studies and the clear implications of not performing surgery after diagnosis of soft tissue sarcoma, inherent selection bias was present in all included studies. Third, moderate to severe heterogeneity was found for most of the outcomes assessed in our study; however, one of the most important study outcomes, which was the association between residual disease and local recurrence, displayed low heterogeneity. Fourth, our funnel plot analysis revealed publication bias, as mild to moderate asymmetry was found in all analyses.

We found no association between a previous UE and worse five-year OS. Our meta-analysis identified three studies that showed better five-year OS in the UE group compared to the PE group [15,18,25]. All three studies, however, reported that tumors in the planned excision group were larger, more often high grade, and deeper than those in the UE group. These findings indicate that the poorer overall survival in the PE group is due to the biological aggressiveness of the tumor, rather than the type of surgery performed. Chandrasekar et al. reported that on univariate analysis, survival was related to tumor grade and size, as well as residual tumor, local recurrence, and excision margins. However, on multivariate analysis only tumor grade was found to be associated [54].

Although the history of UE did not affect overall survival, it did however modify the risk of local recurrence at five years. This could be due to the notion that even when re-excision is performed, the presence of residual disease in tumor bed after an unplanned excision is a risk factor for local recurrence. Since unplanned excisions do not consider surgical margins during resection of the tumor, there is a risk of microscopic or macroscopic persistence of the tumor in the surgical bed. In our study, having residual disease on the tumor bed during re-excision was associated with lower five-year LRFS, with low heterogeneity between all studies included. This result is especially important given that, since residual disease occurs only after unplanned excisions, there is a lesser risk for confounding. The presence of residual tumor during re-excision has been associated with oncologic worse outcomes [6]. Moreover, encountering gross residual disease has been linked with worse LRFS compared to microscopic residual disease: Potter et al. reported five-year LRFS estimates of 94.4%, 61.3% and 0% for patients without residual disease, patients with microscopic residual disease, and patients with gross residual disease, respectively [35].

Our results showed that local recurrence was associated with worse five-year OS. The impact of local recurrence on patient prognosis is a highly debated topic [55,56,57]. While certain authors recommend re-excision after an unplanned excision, other authors advocate a “wait and see approach” under the assumption that local recurrence does not affect prognosis [58]. Studies recommending re-resection report direct correlation between local recurrence and disease-specific survival [59] or distant metastases [60]. However, due to the retrospective nature of these studies, no true cause-effect relationship can be established. Among the studies assessing residual disease we included, we identified a mean positivity rate of almost 60% after re-excision. This means that if a wait and see approach were chosen, over half of all patients would have tumor in situ. Scoccianti et al. gave valuable insight on this topic [41]. They postulated this situation would be difficult to accept for both surgeon and patient, unless evidence proved surgical margins were not a prognostic factor in unplanned excisions, which is contrary to what the literature reflects. Based on our findings, which summarize all available evidence, we consider that re-excision should remain the standard of care after unplanned excision. In addition, efforts should be made to determine the presence of residual disease at the tumor bed. Recent radiology literature has demonstrated the use of MRI to pre-operatively assess for residual tumor [61]. The timing of re-excision procedures should be assessed on an individual basis, taking adjuvant or neo-adjuvant treatments into consideration. Further research should focus on the specific outcomes for microscopic vs. macroscopic residual disease.

Conflicting information in the literature exists regarding the relationship between a previous UE and amputation rates. Traub et al. found a significant difference in amputation rates between UE and PE groups (18.1% vs. 10.1%, respectively) and suggested that tissue contamination from an initial UE could result in the need for more extensive surgery [44]. Other studies, however, found that PE patients were subject to more amputations when compared to subjects exposed to a previous UE. This could be explained by the tendency to have larger, deeper seated tumors in these studies’ PE groups [14,21]. Our analysis found no association between previous exposure to a UE and a higher number of amputations. Ultimately, selection bias is present behind the decision to amputate, as the indication resides in patient and tumor characteristics, as well as the response to neoadjuvant or adjuvant treatments.

Several studies have suggested that UE could lead to more PRS when compared to PE [4,9,12,30,42,44]. Although our analysis was characterized by high study heterogeneity, we failed to see any association between UE and PRS rates. However, the type of reconstruction must also be considered when comparing PRS rates. Tokumoto et al. found that, while superficial reconstructions were similar between groups, deep reconstructions were more common in the PE group (26.1% vs. 7.5%, *p* < 001). This finding aligns with the notion that patients with previous UE have smaller, more superficial masses. On the contrary, two high-sample studies reported otherwise [14,21]. Bateni et al. found the rate of reconstruction in patients with PE that had been subject to an MRI and biopsy to be 29.1%, almost threefold that in the UE group (9.9%). Interestingly, Danielli et al. reported different rates for plastic reconstruction between macroscopically complete and incomplete UE (18.7% vs. 34.9%, respectively). Further studies should further assess this difference to accurately establish an association.

## 5. Conclusions

Our study showed that unplanned excision and residual disease at re-excision were associated with worse local recurrence-free survival. We also found local recurrence to be associated with a poorer overall survival in STS patients. Despite the presence of potential confounding factors, our study demonstrates the impact of unplanned excisions on prognosis. Orthopaedic oncologists should consider performing tumor bed re-excision to prevent local recurrence. Future studies should focus on the impact of microscopic and macroscopic residual disease, evaluate whether certain histologic subtypes are more likely to present residual disease on re-excision, and make efforts to control possible confounders such as tumor size, depth, grade, location, use of neo-adjuvant therapies and others.

## Figures and Tables

**Figure 1 cancers-16-00443-f001:**
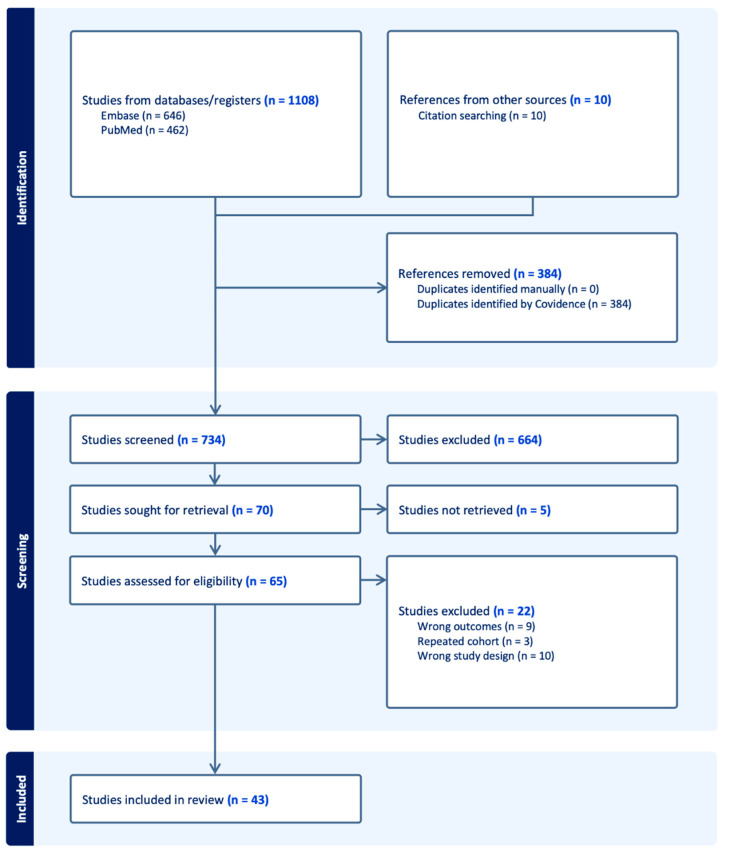
Flowchart of our literature search and selection of relevant articles.

**Figure 2 cancers-16-00443-f002:**
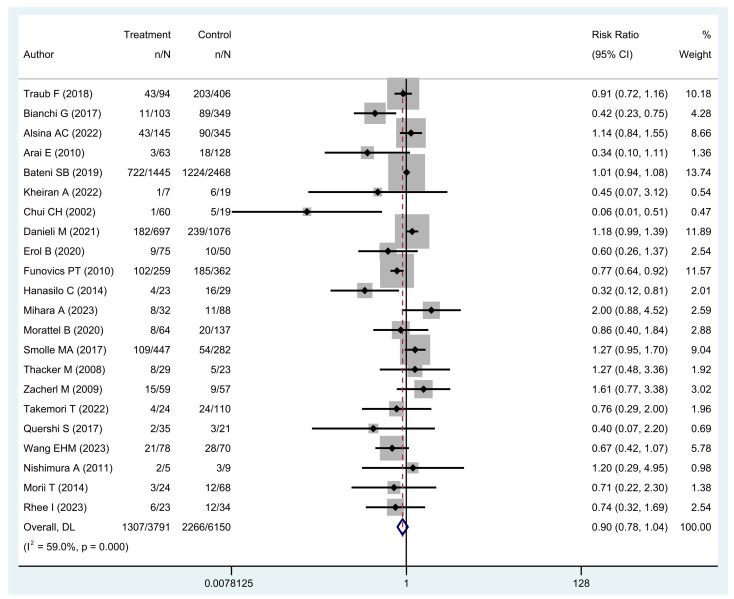
Forest plot of studies assessing risk ratios (RR) for five-year overall survival in patients with unplanned excisions and patients with planned excisions [4,8,10,12,14,15,18,21,23,24,25,26,30,31,32,34,36,39,42,43,44,46].

**Figure 3 cancers-16-00443-f003:**
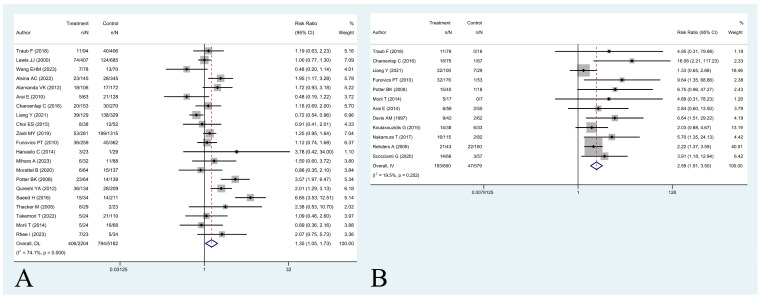
(**A**,**B**) Forest plot of (**A**) studies assessing risk ratios (RR) for five-year local recurrence-free survival in patients with unplanned excisions and patients with planned excisions and (**B**) studies assessing risk ratios (RR) for five-year local recurrence-free survival in patients with positive residual disease at re-excision and patients with negative residual disease at re-excision [3,8,10,11,12,13,16,17,24,25,27,28,29,30,31,32,33,35,37,38,39,40,41,42,43,44,47].

**Figure 4 cancers-16-00443-f004:**
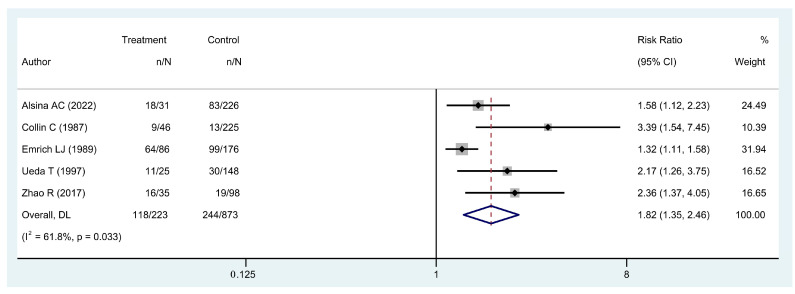
Forest plot of studies assessing risk ratios (RR) for five-year overall survival in patients that developed local recurrence and patients that did not develop local recurrence [8,19,22,45,48].

**Figure 5 cancers-16-00443-f005:**
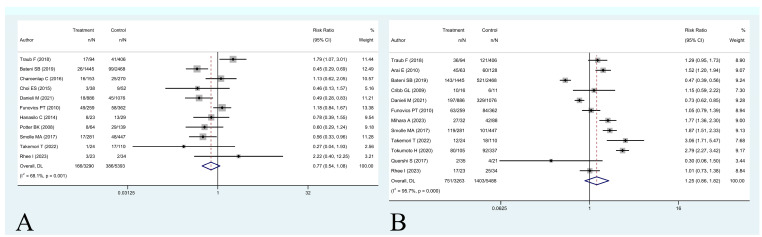
(**A**,**B**) Forest plot of studies assessing risk ratios (RR) for (**A**) amputation rate and (**B**) plastic reconstruction surgery rate in patients with unplanned excisions and patients with planned excisions [4,9,12,14,16,17,20,21,24,25,30,35,36,39,42,44].

**Table 1 cancers-16-00443-t001:** Characteristics of all studies included in the meta-analysis.

Author	Sample (*n*)	PE (*n*)	UE (*n*)	Age (Years) ^e^	Follow-Up (Months) ^e^	Outcomes
Alamanda et al. [11]	278	172	106	58.5 ^+^/55.5 ^+^	37.2 ^+^	5y LRFS
Alsina et al. [8]	490	345	145	50.1 */47.3 *	57.1 */55.8 *	5y OS ^c^, 5y LRFS
Arai et al. [12]	191	128	63	50.5 */51 *	55 *	5y OS, 5y LRFS, PRS
Arai et al. [13] ^b^	113	0	113	-/49 *	69 *	5y LRFS (UE)
Bateni et al. [14]	3913	2468	1445	77.4/77.6	42 ^+^	5y OS, AMP, PRS
Bianchi et al. [15]	452	349	103	59 ^+^/51 ^+^	98 ^+^/46^+^	5y OS
Charoenlap et al. [16]	451	290	161	-	72.6 ^+^	5y LRFS ^a^, AMP
Choi et al. [17]	90	52	38	31.3 */34.7 *	72 *	5y LRFS, AMP
Chui et al. [18]	79	19	60	-	-	5y OS
Collin et al. [19] ^d^	271	-	-	-	98.4 ^+^	5y OS
Cribb et al. [20]	27	11	16	45.9 */59.1 *	90 *	PRS
Danieli et al. [21]	1962	1076	886	59 ^+^/-	85.2 ^+^	5y OS, AMP, PRS
Davis et al. [3] ^b^	104	0	104	-	36.8 *	5y LRFS (UE)
Emrich et al. [22] ^d^	262	-	-	-	64 ^+^	5y OS
Erol et al. [23]	125	75	50	47.2 */47.3 *	-	5y OS
Funovics et al. [24]	621	362	259	49 */53 *	-	5y OS, 5y LRFS ^a^, AMP, PRS
Hanasilo et al. [25]	52	29	23	-	39.9 *	5y OS, 5y LRFS, AMP
Kheiran et al. [26]	26	19	7	54.4 */65.9 *	75.6 *	5y OS
Koulaxouzidis et al. [27] ^b^	71	0	71	-/55.9 ^+^	-	5y LRFS (UE)
Lewis et al. [28]	1092	685	407	-	57.6 ^+^	5y LRFS
Liang et al. [29]	458	329	129	-	112.2 ^+^	5y LRFS ^a^, 5y MFS
Mihara et al. [30]	120	88	32	58.9 */60.9 *	48.3 */50.8 *	5y OS, 5y LRFS, PRS
Morattel et al. [31]	201	137	64	55.9 ^+^/52.5 ^+^	89 */99 *	5y OS, 5y LRFS
Morii et al. [32]	92	68	24	58.1 */61.8 *	60 *	5y OS, 5y LRFS ^a^
Nakamura et al. [33] ^b^	197	0	197	-/54 *	48 *	5y LRFS (UE)
Nishimura et al. [34]	14	9	5	31 */34 *	56.4 *	5y OS
Potter et al. [35]	203	139	64	57 ^+^/53 ^+^	63 */48 *	5y LRFS ^a^, AMP
Quershi et al. [36]	56	21	35	-	-	5y OS, PRS
Qureshi et al. [37]	343	209	134	60 ^+^/56 ^+^	51.6 ^+^	5y LRFS
Rehders et al. [38] ^b^	143	0	143	-/50 ^+^	109 ^+^	5y LRFS (UE)
Rhee et al. [39]	57	34	23	65.7 ^+^/66.2 ^+^	64.7 ^+^	5y OS, 5y LRFS, AMP, PRS
Saeed et al. [40]	245	211	34	57 ^+^/64 ^+^	33.6 ^+^	5y LRFS, AMP
Scoccianti et al. [41] ^b^	125	0	125	-/50 *	130.8 *	5y LRFS (UE)
Smolle et al. [4]	728	447	281	-	-	5y OS, AMP, PRS
Takemori et al. [42]	134	110	24	58 ^+^/61 ^+^	89.5 ^+^/95.5 ^+^	5y OS, 5y LRFS, AMP, PRS
Thacker et al. [43]	52	23	29	-	99 *	5y OS, 5y LRFS
Tokumoto et al. [9]	442	337	105	56.7 */62.1 *	-	PRS
Traub et al. [44]	500	406	94	58.8 */59.4 *	54 */61.6 *	5y OS, 5y LRFS ^a^, AMP, PRS
Ueda et al. [45] ^d^	173	-	-	-	48 ^+^	5y OS
Wang et al. [10]	148	70	78	-	52.8 ^+^	5y OS, 5y LRFS
Zacherl et al. [46]	116	57	59	50.4/57.8	64.7 *	5y OS
Zaidi et al. [47]	1596	1315	281	57.6 */54.9 *	-	5y LRFS
Zhao et al. [48] ^d^	133	-	-	-	68 ^+^	5y OS

AMP: amputation rate; LRFS: local recurrence-free survival; OS: overall survival; PE: planned excision; PRS: plastic reconstructive surgery rate; UE: unplanned excision. Values displayed refer to ^+^ median or * mean depending on the reporting method chosen by the author. ^a^ These studies additionally reported 5-year LRFS in the UE group by residual tumor status. ^b^ These studies only reported 5-year LRFS in the UE group by residual tumor status. ^c^ These studies additionally reported 5-year OS by local recurrence status. ^d^ These studies only reported 5-year OS by local recurrence status. ^e^ The values displayed in this columns refer to patients with planned excision (first number) and with unplanned excision (second number). If a single number is displayed, it refers to the overall cohort.

**Table 2 cancers-16-00443-t002:** Tumor characteristics by group among included studies.

Author	M:F Ratio	Size (cm) ^b^	Deep Tumor (%) ^c^	High Grade (%) ^c^	Final Positive Margin (%) ^c^
Alamanda et al. [11]	1.29/1.1	12 ^+^/5 ^+^	91.9/77.4	75.6/74.5	9.3/5.7
Alsina et al. [8]	-	10.2 */6.2 *	-	86.9/88.3	-
Arai et al. [12]	1.21/1	9 */4.6 *	70/13	62/52	-
Arai et al. [13]	-/0.9	-/4.5 *	-/15.9	-/77	-/3.4
Bateni et al. [14]	1/1.2	-	37.4/23.9	54.6/46.9	-
Bianchi et al. [15]	1.27/1.1	78%/43%	94/72	100/100	3/17
Charoenlap et al. [16]	1.18/1.4	74.5%/42.9%	85.9/53.3	91/73.4	8/5
Choi et al. [17]	1.74/1.5	9.1 */5.2 *	96/71	100/95	13/21
Chui et al. [18]	-	52.6%/21.7%	-	57.9/40	0/0
Collin et al. [19] ^a^	-	-	-	-	-
Cribb et al. [20]	-	-	-	-	-
Danieli et al. [21]	1.19/1.3	9 ^+^/6 ^+^	86.1/47.9	80.8/75.6	-
Davis et al. [3]	-	-	-	-	-
Emrich et al. [22] ^a^	-	-	-	100	-
Erol et al. [23]	0.74/1.3	9.6 ^+^/5.6 ^+^	100/84	82.7/76	-
Funovics et al. [24]	1.23/1.1	-	-	86.5/88.4	10.5/10.8
Hanasilo et al. [25]	1.23/1.6	89.7%/87%	100/78.3	100/82.6	3.4/17.4
Kheiran et al. [26]	0.58/0.8	≥4 cm: 42.1%/≥4 cm: 14.3%	63.2/85.7	84.2/85.7	-
Koulaxouzidis et al. [27]	-/1.2	-/48.2%	-/65.9	-/36.5	-
Lewis et al. [28]	1.24/1.1	74.4%/40.3%	88.2/60.7	67.6/62.7	25.5/9.1
Liang et al. [29]	1.49/1.6	52%/29.5%	62/44.2	67.4/76	-
Mihara et al. [30]	0.91/1.9	8.9 */3.89 *	39/46	78/74	-
Morattel et al. [31]	1.36/1.2	75.9%/39.1%	81/43.8	36.5/37.5	-
Morii et al. [32]	1.06/0.8	-	73/20.8	80/70	13/16
Nakamura et al. [33]	-/1.3	-/4.7 *	-/33	-/68.5	-/3.6
Nishimura et al. [34]	3.5/0.7	-	-	22.2/40	-
Potter et al. [35]	1.14/1.6	11.5 */8.9 *	75/33	100/100	1/6
Quershi et al. [36]	4.25/1.7	54%/34.2%	100/82.8	41.6/77	-
Qureshi et al. [37]	1.3/1.4	87.7%/40.3%	74.2/25.4	64.6/55.3	-
Rehders et al. [38]	-/1.3	-/66%	-	-/72	-
Rhee et al. [39]	1/0.9	70.6%/69.6%	70.6/52.2	100/100	-
Saeed et al. [40]	-	8.5 ^+^/4 ^+^	-	75.5/82.3	-
Scoccianti et al. [41]	-/0.8	-	-	-/72.5	-/6.1
Smolle et al. [4]	0.95/1.3	28%/20%	78.7/49.8	77.4/74.3	-
Takemori et al. [42]	0.9/2.4	7.50 ^+^/3.55 ^+^	78.2/29.2	-	11.8/12.5
Thacker et al. [43]	-	-	-	-	-
Tokumoto et al. [9]	1.13/1.6	-	-	70.6/48.6	-
Traub et al. [44]	1.33/1.3	12.5 */10.3 *	-	100/100	16.3/12.8
Ueda et al. [45] ^a^	1.4	59.5%	65.9	72.3	-
Wang et al. [10]	1.26/1	11 ^+^/8.8 ^+^	87.1/76.9	50/69.2	14.3/21.8
Zacherl et al. [46]	1.59/1	83.6%/72.6%	90.9/51.9	77/76	3.5/1.7
Zaidi et al. [47]	-	10.7 ^+^/4.6 ^+^	91/74	71/74	17/14
Zhao et al. [48] ^a^	1.6	56.4%	-	100	6.8

Values displayed refer to ^+^ median or * mean depending on the reporting method chosen by the author. ^a^ These studies report a mixed population of planned and unplanned excisions. ^b^ Percentages in this column refer to tumors larger than 5 cm. ^c^ The values displayed in these columns refer to patients with planned excision (first number) and with unplanned excision (second number).

**Table 3 cancers-16-00443-t003:** Reported outcomes by studies included in our meta-analysis.

Author	5-Year OS ^a^ (%)	5-Year LRFS ^a^ (%)	PRS Rate ^a^ (%)	AMP Rate ^a^ (%)	5-Year LRFS (UE Group [RD+ vs. RD-]) ^b^ (%)	5-Year OS (LR+ vs. LR-) ^c^ (%)
Alamanda et al. [11]	-	90.2/83	-	-	-	-
Alsina et al. [8]	73.9/70.3	91.9/84.1	-	-	-	42.7/63.2
Arai et al. [12]	85.7/95.7	83.4/92.2	47/71	-	-	-
Arai et al. [13]	-	-	-	-	89/97	-
Bateni et al. [14]	50.4/50	-	30.1/24.2	6.8/4.9	-	-
Bianchi et al. [15]	74.4/89.4	-	-	-	-	-
Charoenlap et al. [16]	-	88.9/86.9	-	9.4/10.7	75.7/98.6	-
Choi et al. [17]	-	77/79	-	17/8	-	-
Chui et al. [18]	73.7/98.2	-	-	-	-	-
Collin et al. [19]	-	-	-	-	-	80/94
Cribb et al. [20]	-	-	54.5/62.5	-	-	-
Danieli et al. [21]	77.8/86.9	-	30.6/22.2	4.2/2	-	-
Davis et al. [3]	-	-	-	-	79.5/97	-
Emrich et al. [22]	-	-	-	-	-	26/44
Erol et al. [23]	88/80	-	-	-	-	-
Funovics et al. [24]	48.8/60.5	87.6/86.1	23.2/24.3	16/18.9	82/98.2	-
Hanasilo et al. [25]	96.5/86.2	46.1/82.7	-	44.8/34.8	-	-
Kheiran et al. [26]	68.4/85.6	-	-	-	-	-
Koulaxouzidis et al. [27]	-	-	-	-	62.5/83.3	-
Lewis et al. [28]	-	81.9/81.9	-	-	-	-
Liang et al. [29]	-	58/70	-	-	68.1/75.9	-
Mihara et al. [30]	87.1/76.5	87.2/79.9	48/84	-	-	-
Morattel et al. [31]	85.4/87.8	89.3/90.6	-	-	-	-
Morii et al. [32]	82/87.8	76.3/79.8	-	-	72.1/100	-
Nakamura et al. [33]	-	-	-	-	86.2/97.3	-
Nishimura et al. [34]	65.6/60	-	-	-	-	-
Potter et al. [35]	-	89.7/63.7	-	21/13	61.3/94.4	-
Quershi et al. [36]	84.9/93.5	-	20.8/6	-	-	-
Qureshi et al. [37]	-	86.8/73.1	-	-	-	-
Rehders et al. [38]	-	-	-	-	50.4/77.7	-
Rhee et al. [39]	64.5/75.9	84.4/70.1	73.5/73.9	5.9/13	-	-
Saeed et al. [40]	-	93.2/56.1	-	0/0	-	-
Scoccianti et al. [41]	-	-	-	-	79.7/94.7	-
Smolle et al. [4]	75.6/81.9	-	22.6/42.3	10.7/6.0	-	-
Takemori et al. [42]	77.9/83.3	81.2/78.1	16.3/0.5	15.4/4.1	-	-
Thacker et al. [43]	80.2/73.4	90.8/79	-	-	-	-
Tokumoto et al. [9]	-	-	27.3/76.2	-	-	-
Traub et al. [44]	50.1/54	90.1/88.3	39.9/56.4	10.1/18.1	85.5/100	-
Ueda et al. [45]	-	-	-	-	-	54.1/79.5
Wang et al. [10]	60/73	82/91	-	-	-	-
Zacherl et al. [46]	84.2/74.5	-	-	-	-	-
Zaidi et al. [47]	-	84.9/81.2	-	-	-	-
Zhao et al. [48]	-	-	-	-	-	53.6/80.5

^a^ The values displayed in these columns refer to patients with planned excision (first number) and with unplanned excision (second number). ^b^ The values displayed in this column refer strictly to patients with unplanned excision with residual disease on tumor bed re-excision (first number) and without residual disease (second number). ^c^ The values displayed in this column refer strictly to patients with local recurrence (first number) and without local recurrence (second number). AMP: amputation; LR: local recurrence; LRFS: local recurrence-free survival; OS: overall survival; PRS: plastic reconstructive surgery rate; RD: residual disease; UE: unplanned excision.

## Data Availability

The original data presented in the study are included in the article (and Appendix A).

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
