# Peer review of "Is Unplanned Excision of Soft Tissue Sarcomas Associated with Worse Oncological Outcomes?—A Systematic Review and Meta-Analysis"

_cancers, 2024, doi:10.3390/cancers16020443_

Round 1
Reviewer 1 Report
Comments and Suggestions for Authors
Generally, the work is interesting and valuable in the field of cancer research, and is of interest to the broad readership of Cancers. This article should be accepted for publication in Cancers. There are some questions, that should be addressed.
Comments:
1. What percentage of additional resection specimens from UE patients show residual tumor on pathology? And is the percentage showing residual tumor related to tumor type?
2. Is there any difference in prognosis between patients who underwent re-resection and those who were followed up in a wait-and-see policy?
Reviewer 2 Report
Comments and Suggestions for Authors
Thank you for allowing the opportunity to review this manuscript. The authors seek to perform a systematic review and meta analysis comparing outcomes in unplanned excisions vs oncology resections. The followed an appropriate PRISMA methodology. Given the rarity of this disease entity, it is difficult to have a sufficiently powered study to answer questions around sarcoma and this review offers an opportunity for more pooled results. As is usually the case, the review is inherently limited by its contributing components which are generally of poor methodologic quality, thus limiting the overall impact of the study. Additionally, the tremendous heterogeneity that exists between comparison groups limits the conclsusions. Despite this, the authors did utilize appropraite methodology. The authors find that UE was associated with LRFS, though these were not associated with a worse rate of OS, amputation rate or plastic surgery rate.
Specific comments below:
1) It should be noted that such a review has been endeavored before:
https://pubmed.ncbi.nlm.nih.gov/34099954/
Please discuss how this current review is sufficiently different to warrant consideration.
Section 2.2- It is unclear to me what the exact inclusion criteria is. I note the mesh terms, but it would be beneficial to state explicitly "Manuscripts that reviewed outcomes following unplanned excision etc." would be beneficial. The entire first paragraph is similarly a bit cluttered and difficult to follow. I think some rewording could make this section significantly easier to follow and clearer.
The authors mention in line 125: "All 26 reported a higher mean or median in the PE 125 group. Out of 24 studies that reported information on tumor grade, 22 reported a higher 126 percentage of deep tumors in the PE group" These are significant confounding factors that should be discussed.
Line 126: This sentence does not make sense: "Out of 24 studies that reported information on tumor grade, 22 reported a higher percentage of deep tumors in the PE group." Please explicitly
Tables 1-3 are very dense. With >40 included studies this is not unexpected. However, I wonder if there is a simpler way to present this data that would be easier to interpret for the readership. Would it make more sense to present the pooled results instead? This would help with the above.
In the abstract, the authors state that "Unplanned excision of Soft Tissue Sarcomas is associated with worse oncologic outcomes." The authors must specify which outcomes here and throughout the manuscript as this is not necessarily supported by their results (i.e. OS was not different)
While the overall methodology was appropriate and the results very comprehensively reported, I again think that the overall conclusions of the study are limited by the population of studies that were included. I think it is difficult to really interpret these results as the authors mention themselves that the group with planned excisions tended to have larger and deeper tumors (and higher grade?). As this is a systematic review and meta analysis there really is not a way to adequately control for these confounders, which is a major limitation. There is not a way to rectify this really, but it is a fundamental weakness of the study as a whole. This is a major issue that the authors must justify as how can we really draw meaningful conclusions when significant confounders exist between groups?
Comments on the Quality of English Language
No specific comments.
Round 2
Reviewer 2 Report
Comments and Suggestions for Authors
Thank you for addressing my comments. I believe the quality of the study is improved as a result. I think the study is well done. The biggest challenge is the inability to appropriately control for important confounding variables, but this is inherent to a systematic review and meta analysis. Nonetheless, the findings are interesting and of relevance to the sarcoma community.